# A Review of Distributed Secondary Control Architectures in Islanded-Inverter-Based Microgrids

Omar F. Rodriguez-Martinez †, Fabio Andrade *, Cesar A. Vega-Penagos † and Adriana C. Luna

Electrical and Computer Engineering Department, University of Puerto Rico at Mayaguez, Mayaguez, PR 00680, USA; omar.rodriguez27@upr.edu (O.F.R.-M.); cesaraugusto.vega@upr.edu (C.A.V.-P.); adriana.luna4@upr.edu (A.C.L.)
* Correspondence: fabio.andrade@upr.edu; Tel.: +1-787-966-5427
† These authors contributed equally to this work.

**Abstract:** The increasing energy demand, the shortage of energy resources, and the environmental challenges faced by conventional power-generation systems are some of the ongoing challenges faced by modern power systems. Therefore, many efforts have been made by the scientific community to develop comprehensive solutions to overcome these issues. For instance, current technological advances have allowed the integration of distributed generators into the power systems, promoting the use of microgrids to overcome these issues. However, the use of renewable distributed generators have introduced new challenges to the traditional control system schemes. To overcome these challenges, a hierarchical control approach has been proposed for distributed renewable sources. In other words, the control scheme have been divided into three hierarchical levels, primary, secondary, and tertiary, to overcome the new challenges present in modern power systems. Due to extensiveness of this topic, this overview is focused on secondary control systems, mainly for AC isolated microgrids. To improve the power quality of modern systems, several secondary control schemes have been proposed to overcome the well-known problem of frequency and voltage deviation. Some of these schemes have also introduced adequate active/reactive power sharing techniques to optimize the utilization of resources. Additionally, other secondary control schemes have also focused on reducing the communication load, to lower the network cost and adding robustness against communication problems. This article presents an insight of the different control techniques used to overcome power quality and communication problems. A comprehensive overview of distributed secondary control techniques for islanded microgrids is presented. In addition, the implementation of these techniques is explained in an orderly and sequential manner.

**Keywords:** decentralized control; distributed control; microgrids; power system; renewable sources; secondary control




## 1. Introduction

A microgrid (MG) is a set of distributed energy resources and co-dependent loads with defined electrical limits that work as a single controllable system with respect to the grid. The voltage, frequency, and active/reactive power are the primary variables used to control the operation of a MG [1]. An MG can be connected and disconnected from the grid, allowing it to operate in isolated mode [2]. In grid-connected MGs, the frequency and the voltage at the point of common coupling (PCC) are dominantly determined by the main grid. Moreover, a power shortage can also be provided by the main grid [3]. In stand-alone mode, the real and reactive power produced in the MG must be balanced depending on the demand of the local loads. This operational mode is more demanding since the frequency and voltage of the MG are controlled by different distributed energy resource (DER) units, instead of the main grid.

The power balance of the system can be achieved using centralized and decentralized control approaches. The centralized control approach is based on a central controller

that assigns the required set point to all the DER units and controllable loads. This approach introduces a high communication load, since constant communication between the central controller and the DER units is required. Conversely, in decentralized control, each unit is governed by its own local controller, and the set points are determined based on local measurements. In other words, each controller is not entirely aware of the status of system-wide variables nor actions taken by other controllers [3–5]. This approach aims to ensure that all units correctly contribute to supply the load in a pre-specified or optimized way. A block diagram of centralized and decentralized secondary control systems is presented in Figure 1.

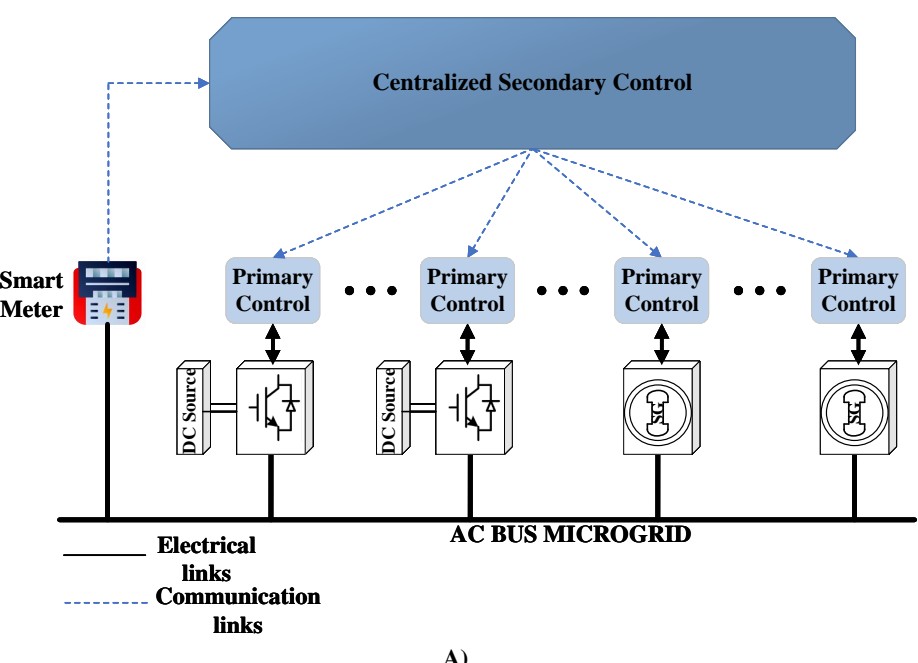

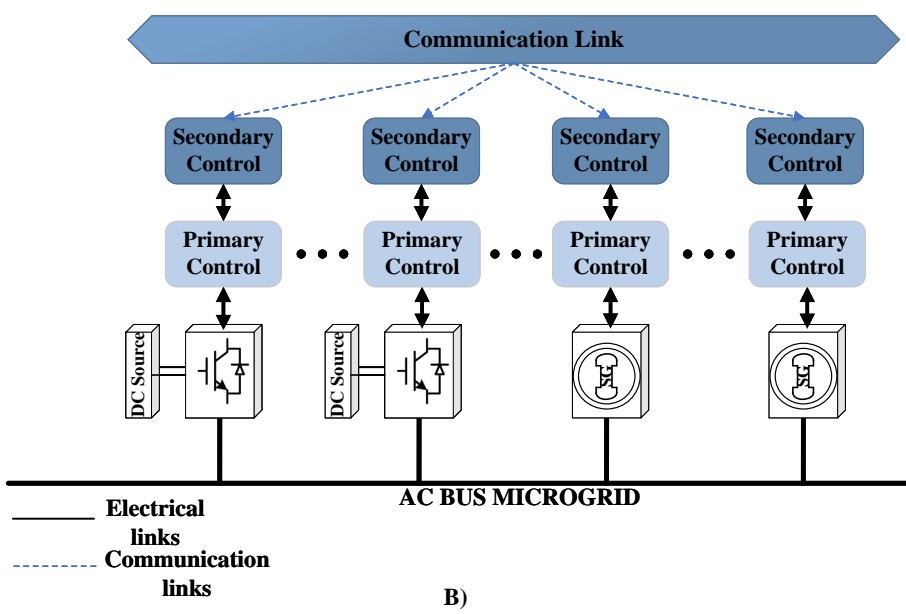

**Figure 1.** Microgrid control architecture with: (**A**) centralized secondary control; (**B**) decentralized secondary control.

The implementation of fully centralized or fully decentralized controllers is not feasible due to the communication and computation load and the strong coupling between the operations of multiple distributed generators (DGs) according to [3]. Therefore, a combination of both types of controllers is necessary through a three-level hierarchical control scheme: primary, secondary, and tertiary [1,6,7]. Each level is different to each other based on the response time, communication condition, and the operational time [3]. However, new approaches have been proposed to unify the hierarchical control scheme [8]. Even though, this review paper provides a brief description of the entire hierarchical control approach, a deeper analysis of the secondary control level is presented.

Primary control is the lowest level in the control hierarchical scheme and it exhibits the shortest response time (milliseconds). At this level, no communication is required and the control is based on local measurements [9]. It is responsible for the voltage and frequency regulation on the MG, applying fast-acting voltage and current inner control loops. Furthermore, due to it fast response time, this control level also supervises the islanding detection, power sharing, and the output power of DG [10–12]. Droop control is the most widely used primary control method, and its implementation emulates the power sharing characteristics of a synchronous generator. Additionally, it is design to work autonomously and provide immediate preset responses to local events [3,13–16]. The limitations of this control level are countered by secondary control.

Tertiary control is the highest level of the control hierarchy. This control level uses the connected loads data, the demand and supply balance, the weather forecast, and the economic dispatch to optimize the electricity cost and improve the reliability among MGs. The output of the tertiary controller is the power output reference used by the secondary controller to make the required adjustments [9,12,13]. Moreover, it provides input signals to other sub-parts in the full grid [3,17,18]. Generally, response time is in the order of several minutes or even hours.

The secondary controller supervises the regulation of the frequency and voltage deviations after the primary controller [1,10,19,20]. Furthermore, it can be used to optimize the active and reactive power sharing within a MG. Compared to the primary control, it has a longer response time, since it is responsible for making corrections after the primary controller has been executed. In addition, it is responsible of keeping the DG variables up-to-date at all times through the communication system [1,19,20].

The secondary control hierarchy can be implemented as centralized, decentralized, and distributed controllers [10,21–26]. The centralized secondary approach uses a master controller with inputs, such as: the data of each DER unit and the load in the MGs, and the forecast information, such as the wind speed, solar irradiance, and local consumption [3]. In this scheme, the frequency and voltage of each DG unit is measured in the master controller and compared with the reference values provided by the grid-connected network. [10]. Some features of this control scheme are voltage control, harmonic cancellation, frequency restoration, and active and reactive power management [27]. This scheme performs very well when the MGs are isolated with a static architecture and critical demand–supply balances. The correct operation of the secondary centralized controller relies on the master controller; thus, any failure in the master controller or the communication structure will negatively impact the system stability and correct operation [27]. Therefore, this control scheme is considered to have low robustness, since the master controller is a common point of failure [28]. The decentralized secondary control seeks to resolve the energy management problem of a MG while offering the highest possible independence for several loads and DER units [3]. Different from centralized control, the decentralized secondary control is implemented locally at each DG. Therefore, the secondary controller can be designed and implemented without remote-based measurement and communication networks. The condition of neighbor DGs is estimated using local measurements [29], allowing new DER units to be included without making changes to the controller configuration. However, it requires a more complex coordination. This type of control is suitable for MG in grid-connected mode with multiple owners and a rapidly changing number of DER units [3].

Similar to the decentralized scheme, distributed secondary control does not need a master controller. The dispersed control effort is distributed along with the distributed MG [30] and all DGs exchange information with each other through digital communications. Therefore, minimal information exchange is required to improve the performance of all units [29]. However, It requires self-sufficient "agents" that cooperate to achieve the objectives [31], making the inter-unit communication crucial to supply control [32]. Hence, it is an excellent alternative because there is little chance of losing the system due to a collapse of a single controller [10]. The implementation of the distributed secondary control has been growing in recent years, since it helps to fulfill ideal dispatch [33], to improve active and reactive power sharing [34], to re-establish frequency and voltage [35], among others. Additionally, the level of trustworthiness and safety during the execution has also been increasing because it offers major benefits compared with the others control approaches [36]. This is a promising option to improve the operation and equilibrium of the MG, while offering other advantages. For instance, resilience to communication failures, and scalability since it allows simpler changes in the MG [27,30,37]. Finally, it exhibits plug-and-play operation of DERs, which is an attractive feature for MGs [28].

Microgrids have exploded in popularity in recent years, particularly in hurricane- and earthquake-prone areas that presents a not very robust and outdated grid [38]. This has generated the need to install microgrids throughout the island in order to provide a reliable and secure service. In the event of a blackout or outage of the main grid, microgrids may operate in isolation, allowing for rapid service recovery and reliability. Furthermore, because of advancements in controllers and energy management systems, it allows a safe and reliable operation. The main contribution of this work is to review and explain different control methods used to implement secondary control. This paper summarizes, in an orderly and sequential manner, the current methods used to overcome the power quality and communication problems generated in inverter-based isolated microgrids, describing the methods used to design and implement the different mathematical formulas to achieve the objective of the different controllers. Sections 2 and 3 show a review of secondary control methods focused on power quality and secondary control methods focused on improved communication robustness and decrease computational burden, respectively. Section 4 show a brief discussion interpreting the results obtained. Finally, Section 5 contains the conclusion of this research.

## 2. Secondary Control Methods for Power Quality

Many secondary control schemes (mostly distributed) have been proposed in isolated microgrids to improve the efficiency of the controllers. The use of these schemes avoids power-quality problems, such as voltage and frequency deviations. Additionally, accurate power sharing in the grid is achieved [21–23,34,39–45]. In this chapter, some of the most relevant proposed schemes will be discussed.

### 2.1. Finite-Time Consensus-Based Approach

This technique has been investigated and implemented by different authors, such as [46–49]. The implementation of this technique will be described below based on two different investigations. In [34], authors present different distributed secondary control techniques. The proposed scheme, ensure frequency and voltage restoration, while maintaining an accurate active and reactive power sharing using a finite-time islanded microgrid. A finite-time consensus-based controller was implemented to correct frequency and active power deviations, whereas an observed-based controller was proposed for voltage/reactive-power compensation. The consensus-based controller was implemented using a combination of the sign function and a fractional power integrator. Moreover, saturation constraints were used to create bounded control inputs, reduce transient over-

shoots, and ensure the steady state stability. The control laws for this system can be represented as:

$$
\begin{cases}
u_i^\omega &= sat_{\delta_\omega}\left[k sig(e_i^\omega)^{\frac{1+\alpha}{2}}\right] + \hat{u}_i \\
\dot{\hat{u}}_i &= sat_{\delta_\omega}\left[\gamma sig(e_i^\omega)^\alpha\right]
\end{cases} \quad i = \{1, \dots, N\} \tag{1}
$$

$$
u_i^p = P_{i,max} sat_{\delta_p}\left[k sig(e_i^p)^\alpha + \gamma e_i^p\right] \quad i = \{1, \dots, N\} \tag{2}
$$

where $\alpha$ is the fractional power constant for finite time integrators, $k$ and $\gamma$ are the control gains, and $\delta_\omega$ and $\delta_p$ are the saturation constant for active power and frequency control signals. N is the *ith* DG, $P_{i,max}$ is the maximum active power of the *ith* DG, and $u_i^p$ and $u_i^\omega$ are the control actions to recover frequency and power. Control laws (1) and (2) depend on the frequency and the active power sharing state errors of each distributed generator. The state errors can be modeled as:

$$
\begin{cases}
e_i^\omega &= \sum\limits_{j \in N_i^\omega} a_{ij}^\omega \left(\omega_j - \omega_i\right) + b_i^\omega \left(\omega^{ref} - \omega_i\right) \\
e_i^p &= \sum\limits_{j \in N_i^p} a_{ij}^p \left(\frac{P_j}{P_{j,max}} - \frac{P_i}{P_{i,max}}\right)
\end{cases} \tag{3}
$$

The state errors move through the communication network implemented in the system. Therefore, for the correct functioning of the controller, the communication network was modeled as a digraph $G(V, \varepsilon, A)$. In the light of that, the digraph is composed by a node set of $V = \{V_1, \dots, V_N\}$ representing all the DGs in the MG, a set of edges $\varepsilon \subseteq V \times V$ representing the communication links within the MG and a weighted adjacency matrix $A = (a_{ij})_{(N \times N)}$. The cyber-network of frequency and active power must contain a spanning directed tree, which means that there is a root node that have a direct path to all other nodes on the graph. The microgrid was considered as a multi-agent system in which each DG work as a follower-agent. As a result, the reference values must be available for at least one of the DGs to ensure the correct operation of the system. If one DG has these data, the other ones can access it using the information exchange between generators. By using this scheme, a reduction in the inherent frequency and voltage coupling is achieved for non-uniform line impedances. The coupling reduction allows the independent design of voltage and frequency controllers.

For the implementation of the voltage observed-based controller in (4) and (5), a consensus-based algorithm was used to estimate the voltage of each DG. Subsequently, using a leader–follower-based pinning control mechanism, the derived estimation for each DG was asymptotically pinned to the required reference value. The voltage estimation and control laws for this controller can be expressed as:

$$
\hat{v}_i(t) = v_i(t_0) + \int_{t_0}^t sat_{\delta_v}\left[\dot{v}_i(\tau) + \sum_{j \in N_i^v} a_{ij}^v(\hat{v}(\tau) - \hat{v}(\tau))\right] d\tau \tag{4}
$$

$$
\begin{cases}
u_i^v &= b_i^v\left(v^{ref} - \hat{v}_i\right) + e_i^q \\
u_i^q &= Q_{i,max} sat_{\delta_q}\left(e_i^q\right)
\end{cases} \tag{5}
$$

where $A^v = (a_{ij}^v)_{(N \times N)}$ is the weighted voltage adjacency matrix; $\hat{v}_i(t)$ is the estimated voltage of agent $i$ and $\delta_v$ is a constant voltage saturation. Similarly, to the frequency and active power control laws, (5) depends on the reactive power error, which is used as a constraint to assure accurate reactive power sharing as:

$$
e_i^q = \sum_{j \in N_i^q} a_{ij}^q\left(\frac{Q_j}{Q_{j,max}} - \frac{Q_i}{Q_{i,max}}\right) \tag{6}
$$

This error travels through the communication network of the system. Therefore, to achieve a good controller performance, the network also needs to be strongly connected, which means that a directed path must exist between any pair of nodes in the graph. Figure 2 shows a graphical summary of the proposed approach.

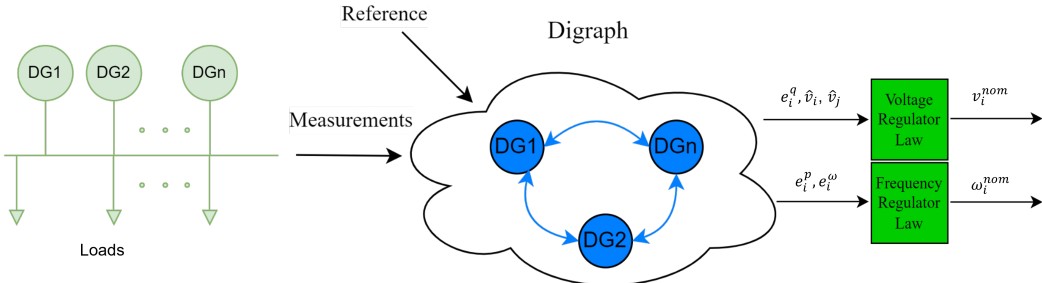

**Figure 2.** Diagram of the implemented secondary consensus-based controller.

To validate the performance of the proposed scheme, six different scenarios were used: load variation, plug-and-play capability, communication topology change, link failure, communication delays, and data drop-out. After analyzing the results obtained in each scenario, it was confirmed that the proposed system is functional despite links loss, data drop-out or communication delays. In addition, the validation results showed that the proposed frequency and active power-sharing control can be achieved in finite time, which allows the voltage and reactive power-sharing control to operate on a slow time scale. Furthermore, unlike asymptotic convergence schemes, this approach allows partial reduction in the inherent voltage and frequency coupling for the case of non-uniform line impedances. It is important to notice that unlike previous works, this paper considers reactive power sharing restrictions. Finally, it was verified that the proposed system can guarantee the data security and its own stability through a key management controller (KMC) [34].

The authors in [41] proposed distributed primary and secondary control schemes for single and three phase microgrids (S/T-MGs). For this study, unbalanced distributed generators and loads non-idealities were considered. As in [34], the communication network was modeled as a digraph and the microgrid was considered as a multi-agent system, where each DG is a follower-agent. Moreover, the distributed secondary control was based on a consensus problem. However, in this work, energy storage systems (ESS), renewable energy sources (RES), and communication delays constraints were taken into consideration. For the primary controller, a variation of the virtual synchronous generator control approach was implemented to have independent and flexible control of the active/reactive power and the voltage magnitude of each phase. This flexibility facilitates the design and implementation of the secondary control system in MGs with single-phase and three-phase generators. The proposed phase-independent virtual synchronous generator (P-VSG) control for the primary controller is giving by:

$$\dot{\theta}_{i,b} = \omega_{i,b} = \omega^* + \sum_{b=a,b,c} \Delta\omega_{i,b} \tag{7}$$

$$M_i \Delta\dot{\omega}_{i,b} = P_{RES,i,b} + P_{ESS,i,b} - P_{i,b} - D_{p,i}\Delta\omega_{i,b} \tag{8}$$

$$K_i \dot{E}_{i,b} = Q_{set,i,b} + \Delta Q_{i,b} - Q_{i,b} - D_{q,i}(E_{i,b} - E^* - \Delta E_{i,b}) \tag{9}$$

where $\omega^*$ is the desired angular frequency of the DG, $\omega_{i,b}$ and $E_{i,b}$ are the output angular frequency and phase voltage magnitude of the DG, respectively; $P_{i,b}$ and $Q_{i,b}$ are the active and reactive output powers of each phase; $Q_{set,i,b}$ is the reactive power set-point for phase-b; $P_{RES,i,b}$ and $P_{ESS,i,b}$ are the RES and ESS output power used for phase-b; $M_i$ and $D_{p,i}$ are the virtual inertia and damping constants, and $K_i$ and $D_{q,i}$ are the integrator gain used to regulate the field excitation and the voltage droop coefficient, respectively. By implementing

this controller, a balanced phase shift is obtained. For the single-phase distributed generators (SDG) case, there is just one frequency deviation, thus (7) can be rewritten as:

$$\dot{\theta}_{i,b} = \omega^* + \Delta\omega_{i,b} \tag{10}$$

The proposed secondary control system was designed modeling the microgrid as a consensus control problem of multi-agents, where the uncertain disturbances from the RES, and ESS, and the communication delays constraints were considered. The ESS constraints consider the state of charge (SOC) and the charging/discharging power values of the ESS. Therefore, three different sub-controllers were designed to regulate the active and reactive power, and the voltage deviations. The voltage regulator was divided into two parts, the first part is the secondary voltage regulator, which is in charge of setting each distributed generator voltage to an admissible range. The second part is the voltage unbalance factor (VUF), which oversees the voltage unbalance between the phases, for the three-phase distributed generators (TDG). Therefore, the three implemented sub-controllers can be represented as follows:

$$\begin{cases} u_{i,b}^P(k) &= S_{Vi}\big[P_{ESS,i,b}(k) - p_i P_{ESS,i,b}(k) + \pi_{i,b}(k)\big] \\ \pi_{i,b}(k) &= \sum_{j \in N_i} a_{ij}\Big[x_{p,j,b}\big(k - \tau_{ij}\big) - x_{p,i,b}(k)\Big] - c\Big[x_{p,i,b}(k) - P_{Y_P}\big(x_{p,i,b}(k)\big)\Big] \end{cases} \tag{11}$$

$$u_{i,b}^Q(k) = \sum_{j \in N_i} a_{ij}\Big[x_{q,j,b}\big(k - \tau_{ij}\big) - x_{q,i,b}(k)\Big] - c\Big[x_{q,i,b}(k) - P_{Y_Q}\big(x_{q,i,b}(k)\big)\Big] \tag{12}$$

$$u_{i,b}^E(k) = \sum_{j \in N_i} a_{ij}\Big[E_{j,b}\big(k - \tau_{ij}\big) - E_{i,b}(k)\Big] - c\Big[E_{i,b}(k) - P_{Y_E}(E_{i,b}(k))\Big] \tag{13}$$

$$u_{i,b}^{VUF}(k) = \sum_{j \in N_i} a_{ij}\Big[x_{j,b}^{VUF}\big(k - \tau_{ij}\big) - x_{i,b}^{VUF}(k)\Big] - c\Big[x_{i,b}^{VUF}(k) - P_{Y_E}\big(x_{i,b}^{VUF}(k)\big)\Big] \tag{14}$$

where $k$ is time, $\tau_{ij} < \tau_{max}$ is the communication delay between agents $j$ and $i$; $p_i$ is the feedback damping gain of follower $i$, and $c$ is the pinning gain that represents the communication link between agent $i$ and at least one leader. Equations (11)–(14) represent the active and reactive power sharing regulators, and the voltage and VUF regulators, respectively. It is important to notice that the voltage and VUF regulators were designed considering admissible ranges based on the IEEE 1547 standard. Figure 3 shows a graphical summary of the proposed approach.

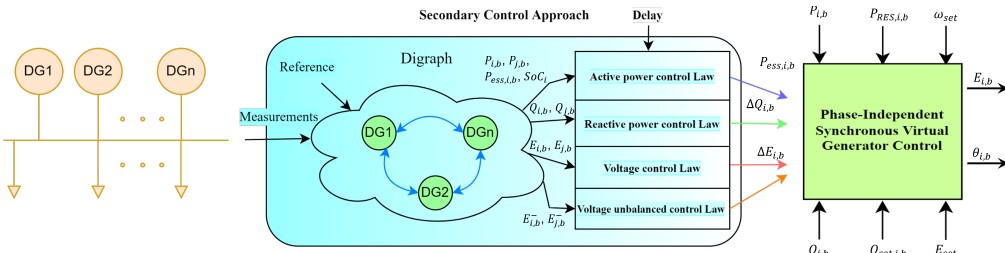

**Figure 3.** Diagram of the implemented secondary consensus-based and the primary phase-independent virtual generator controllers.

The performance of the implemented distributed secondary control scheme was validated using different testing scenarios (load changes and plug-and-play (PnP) capabilities, and communication lost and highly phase unbalanced). Unlike the conventional approach, unbalanced DGs and loads were considered with an appropriate coordination between SDGs and TDGs. Moreover, different from the typical schemes [50–52], the authors implemented a P-VSG control that improves the power control and voltage regulation for each phase, along with the accuracy of the balanced phase shift. The validation results confirmed that the proposed system improved the three-phase balancing performance of DG's (TDGs) (with the drawback of compromising the power sharing among DGs), the

stability and resiliency of the TDGs and SDGs, and the TDG's neutral current reduction. Additionally, to improve the performance during communication lost, the output power of the DGs remained constant (due to the integrator). Then, when the communication is recovered, the system goes back to its normal state.

*2.2. Predictive Model Approach*

This is another technique used to develop secondary controls that has been investigated and implemented by authors, such as [53–56]. As with the previous technique, its implementation will be described with two different investigations. In [44], the authors proposed a distributed secondary control scheme based on a model predictive control system (MPC). The approach proposed by the authors achieved voltage and frequency restoration, ensuring precise active power sharing for islanded microgrids. The sparse communication network described in [34,41] was adopted by the authors in [43]. The proposed approach was divided into two parts, the voltage and frequency control systems. The voltage restoration control system was analyzed as a tracker consensus problem, using an MPC to estimate an appropriate control input $u_i$. The frequency control proposed by the authors, is based on a distributed proportional-integral (PI) method, combined with a finite time observer that provides the system with accurate active power sharing. To enable the independent design of the voltage and frequency controllers, a finite-time voltage observer was used to synchronize the voltage of each distributed generator with the reference voltage. It is important to notice, that the frequency controller can be designed using a constant value of voltage. Therefore, the estimated reference voltage of each DG can be expressed as:

$$\hat{v}_i = sig \left[ \sum_{j=1}^{N} a_{ij} \left( \hat{v}_j - \hat{v}_i \right) + g_i \left( v_{ref} - \hat{v}_i \right) \right]^{1/2} \tag{15}$$

where $g_i$ is the pinning gain representing the immediate access to the reference value. Having the estimated reference voltage, the model for the communication network, and the current state of the system, the evolution of the system for the proposed control approach can model as:

$$\begin{cases} V(k+1) & = & Av(k) + BU(k) + E_r \\ U(k) & = & -FLv(k) + M \end{cases} \tag{16}$$

where $v(k)$ is a vector containing the voltage of all DGs; $U(k)$ is an auxiliary predictive vector that depends on the prediction coefficient $\mu$ and the control horizons steps ($H_u$) to adaptively remove the discrepancies among DG units and the references; $E_r$ is a N-dimensional vector with the reference voltage value that depends on the information update interval($\varepsilon$) and $A, B, F, M$ are the update coefficient matrices. As shown in (16), the use of predictive models transformed the secondary voltage control into a tracker synchronization problem. It can be solved by using the receding-horizon optimization index method [44,45]. Therefore, the receding-horizon optimization index can be defined as:

$$MinJ(k) = \|\Delta V(k+1)\|_Q^2 + \|V(k+1) - \xi I_{NHP}\|_W^2 + \|U(k)\|_R^2 \tag{17}$$

where $Q$, $W$, and $R$ are symmetric, positive and compatible real defined weighting matrices. By using (17), the local neighbor voltage disagreement disappears and the solution converges to the reference value [44,45]. In addition, it allows the computation of the feedback coefficient $\mu$. By replacing $\mu$ to solve (16), the vector $U(k)$ containing $(H_u)$ control steps can be obtained. Therefore, the secondary voltage adjustment ($u_i^V$), can be calculated for each distributed generator as:

$$u_i^V(k) = (\varepsilon + \mu) \left[ - \sum_{j=1}^{N} a_{ij} \left( v_i(k) - v_j(k) \right) - \left( v_i(k) - v_{ref} \right) \right] \tag{18}$$

Therefore, only the first control step $u(k)$ is fed into the system, and the optimization process is carried out again for the time $k + 1$ to achieve a rolling optimization.

For the frequency controller, a frequency observer similar to the one shown in (15) is proposed, and can be modeled as follows:

$$\hat{\omega}_i = sig \left[ \sum_{j=0}^{N} a_{ij} (\hat{\omega}_j - \hat{\omega}_i) + g_i \left( \omega_{ref} - \hat{\omega}_i \right) \right]^{1/2} \tag{19}$$

By using the frequency observer of (19), the frequency control law can be written as a set of two parts. The first one, expressing the tracking of the reference value, and the second one in charge of maintaining an accurate power sharing. Therefore, the frequency control law can be written as:

$$\begin{cases} u_i^{\omega} = & e_i^{\omega} + e_i^{u} \\ \dot{e}_i^{\omega} = & \alpha(\hat{\omega}_i - \omega_i) \\ \dot{e}_i^{u} = & sig \left( \sum_{j=0}^{N} a_{ij} \left( u_j^{\omega} - u_i^{\omega} \right) \right)^{\beta} \end{cases} \tag{20}$$

where $\alpha$ and $\beta$ represents the positive proportional gains of the controller. Figure 4 shows a graphical summary of the proposed approach.

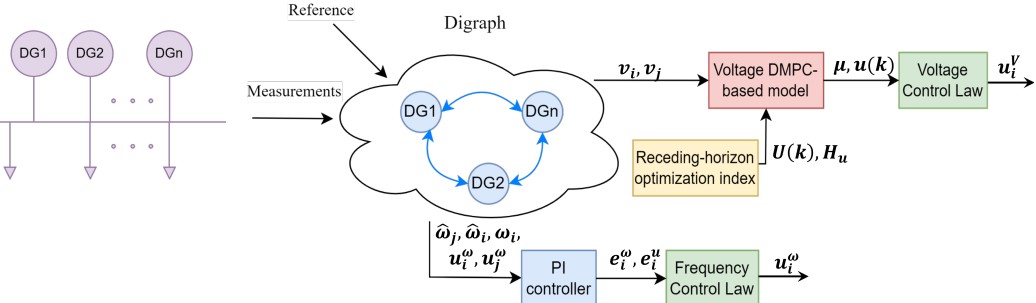

**Figure 4.** Diagram of the implemented secondary predictive controller.

Simulation results validated the robustness of the proposed approach under some specific conditions, such as load disturbances, time-vary communication topologies, parameter perturbations, plug-and-play operation, and variations in the information update interval. The proposed distributed approach removes the requirement for a centralized controller, as each DG only exchanges information with its neighboring DGs. Therefore, this method guarantees that the voltage and frequency of the system are re-established in a drastic accelerated manner, maintaining an accurate power sharing. Due to the rolling optimization used in this approach, the distributed controller can overcome load, communication, and parameter disturbances. The proposed scheme also offers plug-and-play operation within minimal transients and fast dynamics, along with robustness against information update rate changes.

In [45], an improved version of this work was introduced by the authors. Even though the same methodology and methods were used as compared to [44], in this research the authors' introduced the non-linear dynamics of heterogeneous distributed generators to the analysis. The large-signal dynamic model for the DGs was presented as a multiple-input multiple-output (MIMO) system. The proposed system took into consideration the voltage, current, power controllers, the inductance-capacitance (LC) filter, the resistive-inductive

(RL) output connector impedance, and the secondary voltage control signal. The MIMO non-linear system can model as:

$$\begin{cases} \dot{x}_i = f_i(x_i) + k_i(x_i)D_i + g_{i1}(x_i)u_{i1} + g_{i2}(x_i)u_{i2} \\ \qquad\quad y_{i1} = h_{i1}(x_i) = v_{odi} \\ \qquad\quad y_{i2} = h_{i2}(x_i) = \omega_i \end{cases} \tag{21}$$

where $u_{i1} = u_{vi}$ and $u_{i2} = u_{\omega i}$ are the control laws of the secondary control, $y_{i1}$ and $y_{i2}$ are the filtered output voltage and the angular frequency of the *ith* DG. As in [44], this can be analyzed as tracker synchronization, where the appropriate control input $u_i$ is needed for the secondary control system. To simplify the design process of the proposed secondary voltage controller, an input–output linearization (IOFL) method was used to partially linearized the dynamics of the DGs, avoiding the linearization around a prior steady-state operating point. The discrete-time model using the Euler discretization of the distributed MPC can be represented in state variables as follows:

$$\begin{cases} z_{iv}(k+1) = A_{id}z_{iv}(k) + B_{id}v_i(k) \\ \qquad\quad y_{i1}(k) = C_{id}z_{iv}(k) \end{cases} \tag{22}$$

where $z_{iv}(k)$ is the current state vector of the system; $y_{i1}(k)$ is the system's output vector; $v_i(k)$ is the input vector of the system, and $A_{id}$, $B_{id}$, and $C_{id}$ are the system matrices. It is important to notice that $H_p$ and $H_u$ will be used as the prediction and control horizon steps. Therefore, having the available current states $z_{iv}(k)$, the control effect, and the discrete system model shown in (22), the futures output values of vector $y_{i1}(k)$ as a function of the incremental control sequence $(\Delta V_i(k, H_u \mid k))$ can be written as:

$$\begin{cases} Y_{i1}(k+1, H_p \mid k) = N_i \Delta V_i(k, H_u \mid k) + M_i(k) \\ M_i(k) = F_i z_{iv}(k) + G_i \Gamma'_i v_i(k-1) \\ \Gamma'_i = \begin{bmatrix} 1 \\ \vdots \\ 1 \end{bmatrix} ; \bar{\Gamma}_i = \begin{bmatrix} 1 & \cdots & 0 \\ \vdots & \ddots & \vdots \\ 1 & \cdots & 1 \end{bmatrix} \end{cases} \tag{23}$$

where $F_i$, $G_i$ and $N_i$ can be written as:

$$\begin{cases} F_i = \begin{bmatrix} C_{id}A_{id} & C_{id}A_{id}^2 \cdots C_{id}A_{id}^{H_P} \end{bmatrix}^T \\[2em] G_i = \begin{bmatrix} C_{id}B_{id} & & \\ C_{id}A_{id}B_{id} & C_{id}B_{id} & \\ \vdots & \vdots & \ddots \\ C_{id}A_{id}^{H_P-1}B_{id} & C_{id}A_{id}^{H_P-2}B_{id} & \cdots & C_{id}A_{id}^{H_P-H_U}B_{id} \end{bmatrix} \\[2em] N_i = G_i\bar{\Gamma}_i \end{cases}$$

Considering the futures output values of $y_{i1}(k)$ in the first term of (23), the receding-horizon optimization index method used to solve the tracker consensus problem can be formulated as follows:

$$\begin{aligned} MinJ_i = & \left\| \frac{1}{|N_i|} \sum_{j \in N_i} Y_j(k+1, H_p \mid k) - Y_i(k+1, H_p \mid k) \right\|_{Q_i}^2 \\ & + \left\| Y_r(k) - Y_i(k+1, H_p \mid k) \right\|_{W_i}^2 + \left\| \Delta V_i(k, H_U \mid k) \right\|_{R_i}^2 \end{aligned} \tag{24}$$

Following the receding horizon principle, just the first term of the rolling optimization sequence of (24) is used to calculate the ongoing control effort as shown below:

$$v_i(k) = v_i(k-1) + \Gamma_i H_i^{-1} N_i^T \Bigg[ W_i(M_i(k) - Y_r(k)) +$$

$$Q_i \left( M_i(k) - \frac{1}{|N_i|} \sum_{j \in N_i} Y_j(k+1, H_p \mid k) \right) \Bigg] \quad (25)$$

where $|N_i|$ is the number of neighbors of the *ith* DG; $W_i$ and $H_i$ are positive definite symmetric weighting matrices, and $Y_r(k)$ is the desired output value during the predictive horizon. Using the control effort value obtained in (25) and the second derivative of output ($y_{i1}$), the voltage control law can be deduced as:

$$u_{vi} = \left( L_{gi1} L_{Fi} h_{i1} \right)^{-1} \left( -L_{Fi}^2 h_{i1} + v_i \right) \quad (26)$$

where $L_{Fi}$ is the inductances of the filter and $h_{i1}$ represents the filtered voltage of the DGs. The value of the voltage control law calculated in (26) is sent to the primary controller of each DG to regulate the voltage deviations. For the PI frequency control, an IOFL was performed to find a correlation between the angular frequency of the *ith* DG ($y_{i2}$) and the frequency control law ($u_{i2}$) Then, the secondary frequency tracker synchronization control can be stated as follows:

$$\begin{cases} u_{\omega i} = \int (e_{\omega i} + e_{Pi}) \\ e_{\omega i} = c_\omega \sum_{j \in N_i} a_{ij}(\omega_j - \omega_i) + g_i(\omega_{ref} - \omega_i) \\ e_{Pi} = c_P sig\left[\sum_{j \in N_i} a_{ij}(m_{Pj}P_j - m_{Pi}P_i)\right]^\alpha \end{cases} \quad (27)$$

It can be seen that the frequency control law is composed of two sections. The first one expresses that all DGs operate as a group to follow the reference value through a sparse communication network, and the second one in charge of maintaining an accurate power sharing in a finite time. Figure 5 shows a graphical summary of the proposed approach.

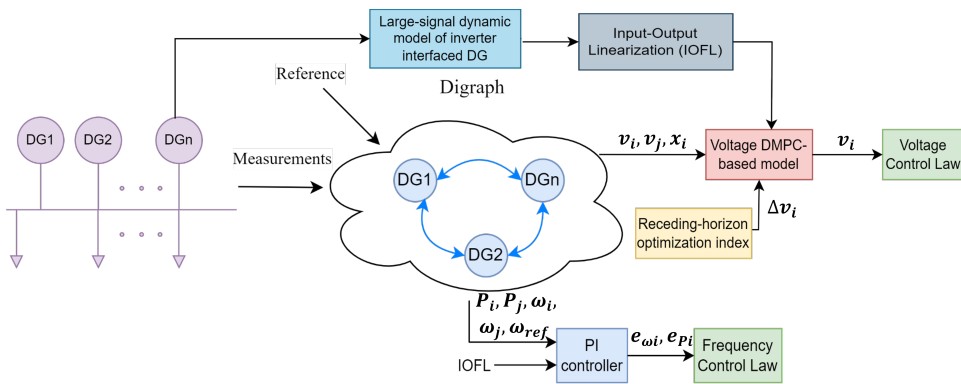

**Figure 5.** Diagram of the implemented secondary predictive controller based on an input–output linearization of the DG inverter model.

In this work, a distributed scheme that provides enhanced flexibility and reliability compared to the centralized approach was proposed. Furthermore, it resolves some uncertainties in the communication links, time delays, and model parameters. The efficiency of the proposed scheme was verified in simulations. For this purpose, different parameters were taken into consideration, such as controller performance, robustness against parameter perturbation and time-varying communication topology, and communication delays. Additionally, the obtained results were compared with the ones obtained using cooperative control method proposed in [57]. The simulation results showed the frequency and voltage

robustness against load disturbances, which just caused minor transients. Additionally, these parameters were successfully held, allowing them to return to their reference value in few seconds, without affecting the accurate active power sharing. Regardless of whether the power demand fluctuates, or the secondary control is implemented, the accurate real power sharing was achieved during the entire runtime. This approach also exhibits faster dynamics compared to the cooperative control method and can reduce the convergence time by including a predictive mechanism. Due to the implemented distributed model predictive control (DMPC), that mitigates the oscillations generated by the delays, the system achieved a desirable performance in pretense of communication networks, system parameters, and load perturbations.

All the communication robustness control strategies discussed are summarized in Table 1. It includes the technique used, active and reactive power sharing, voltage and frequency regulation, method used for stability analysis, demand response method, Harmonic compensation, and unbalance compensation.

**Table 1.** Summary of the recently published secondary control for power-quality compensation.

| Concept | Power Quality Articles | | | | | | | |
|:---:|:---:|:---:|:---:|:---:|:---:|:---:|:---:|:---:|
| | [34] | [39] | [40] | [41] | [42] | [43] | [44] | [45] |
| A | ✓ | ✓ | ✓ | ✓ | ✓ | ✓ | ✓ | ✓ |
| B | ✓ | ✓ | ✗ | ✓ | ✓ | ✓ | ✓ | ✓ |
| C | ✓ | ✓ | ✗ | ✓ | ✓ | ✓ | ✓ | ✓ |
| D | ✓ | ✗ | ✗ | ✓ | ✓ | ✗ | ✗ | ✗ |
| E | ✗ | ✗ | ✓ | ✗ | ✗ | ✗ | ✗ | ✗ |
| F | ✓ | ✗ | ✗ | ✓ | ✗ | ✗ | ✗ | ✗ |
| G | ✗ | ✓ | ✗ | ✗ | ✗ | ✗ | ✓ | ✓ |
| H | ✗ | ✗ | ✗ | ✗ | ✗ | ✓ | ✗ | ✗ |
| I | ✗ | ✗ | ✗ | ✗ | ✓ | ✗ | ✗ | ✗ |
| J | ✓ | ✓ | ✗ | ✓ | ✗ | ✗ | ✗ | ✗ |
| K | ✗ | ✗ | ✗ | ✗ | ✗ | ✗ | ✓ | ✓ |
| L | ✗ | ✗ | ✓ | ✗ | ✓ | ✓ | ✗ | ✗ |
| M | ✗ | ✗ | ✓ | ✗ | ✗ | ✗ | ✗ | ✗ |
| N | ✗ | ✗ | ✗ | ✓ | ✗ | ✗ | ✗ | ✗ |
| O | ✗ | ✗ | ✗ | ✓ | ✗ | ✗ | ✗ | ✗ |
| P | ✓ | ✓ | ✓ | ✓ | ✗ | ✗ | ✓ | ✓ |
| Q | ✗ | ✗ | ✗ | ✗ | ✓ | ✓ | ✗ | ✗ |

Legend: A. Frequency regulation; B. Voltage Regulation; C. Active Power Regulation; D. Reactive Power Regulation; E. Linear Active Disturbance Rejection Control (LADRC); F. Consensus Control Method; G. Prediction Control Method; H. State of Charge Control Method; I. Power Flow Control Method; J. Lyapunov Stability Analysis; K. Lemma Stability Analysis; L. Root Locus Analysis; M. Demand Response; N. Harmonic Compensation; O. Unbalance Compensation; P. Distributed Control; Q. Centralized.

## 3. Secondary Control Methods for Communication Robustness

As discussed in Section 2, there are many challenges in the implementation of control systems (e.g., communication challenges) for MGs, especially for parallel operation of clusters. In a control system, the communication approach can be classified as communication-based or communication-less and this can vary at depending on the layer. In general, the communication network is facing problems at the different layers. For instance, communication latency, data drop-up, and expense issues [58]. Therefore, many research efforts have been made by the scientific community to improve the robustness of the controllers

against communication problems [29,59–66]. In this chapter, different solutions proposed by the scientific community will be discussed.

### 3.1. Layers Coordination Approach

For communication problems, this technique has been investigated as a way to introduce robustness to the communication system [67–69]. The implementation of this technique will be explained by the following two investigations. In [60], a new network configuration scheme was proposed. This configuration allows an adequate voltage and frequency correction, assuring an accurate active and in some cases reactive power sharing. Lu et al., proposed a distributed hierarchical cooperative (DHC) control technique, which is based on a pinning control mechanism from the leader-follower-based multi-agent control theory was proposed. The DHC strategy consist in a two-layer intermittent communication network, for an island-base microgrids cluster. The proposed approach can drive the frequencies/voltages to the reference values for all the distributed generators on the microgrid. At the same time, it provides accurate active and reactive power sharing among microgrids. To achieve this, the authors proposed a two-layer sparse communication network composed by a lower and upper network. The lower network ($\mathcal{G}_s$) was modeled using a pinning-based distributed secondary control scheme, responsible of the frequency/voltage control for each microgrid. The upper network ($\tilde{\mathcal{G}}$) was created by pinning one or some distributed generators form the lower network of each microgrid. For both layers, a previous communication network protocol was adopted [34,39,41,44,45,57].

The upper network was implemented as a consensus-based controller. This controller behaves as a tertiary control and is responsible of the active/reactive power control among microgrids. The distributed secondary control in the lower network, uses a voltage observer to make the voltage of each DG converge to the weighted average values within the MGs, thus, allowing an accurate reactive power sharing. Therefore, the proposed distributed tertiary control (DTC) scheme was designed to regulate the power flow between MGs, balancing their output power. Therefore, assuming that just one DG for each MG is pinned to the tertiary layer, the consensus-based DTC can be modeled as:

$$
\begin{cases}
P_{s,pin}\left(t_k^{\tilde{\ell}+1}\right) = P_{s,pin}\left(t_k^{\tilde{\ell}}\right) + \tilde{u}_{s,pin}^P(k) \\
Q_{s,pin}\left(t_k^{\tilde{\ell}+1}\right) = Q_{s,pin}\left(t_k^{\tilde{\ell}}\right) + \tilde{u}_{s,pin}^Q(k)
\end{cases}
\tag{28}
$$

where $P_{s,pin}$ and $Q_{s,pin}$ are the active and reactive output power of the DTC, $\tilde{u}_{s,pin}^P$ and $\tilde{u}_{s,pin}^Q$ are the tertiary control inputs of the *pinth* DG. These tertiary discrete time control inputs can be designed as:

$$
\begin{cases}
\tilde{u}_{s,pin}^P(k+1) = \sum_{\tilde{k}\in\tilde{N}_s} \tilde{\gamma}_{s\tilde{k}}\tilde{a}_{s\tilde{k}}\left[K_{\tilde{k},pin}^P P_{\tilde{k},pin}(t_k^{T^*}) - K_{s,pin}^P P_{s,pin}(t_k^{T^*})\right]/K_{s,pin}^P \\
\tilde{u}_{s,pin}^Q(k+1) = \sum_{\tilde{k}\in\tilde{N}_s} \tilde{\gamma}_{s\tilde{k}}\tilde{a}_{s\tilde{k}}\left[K_{\tilde{k},pin}^Q Q_{\tilde{k},pin}(t_k^{T^*}) - K_{s,pin}^Q Q_{s,pin}(t_k^{T^*})\right]/K_{s,pin}^Q
\end{cases}
\tag{29}
$$

where $K_{s,pin}^P$ and $K_{s,pin}^Q$ are the droop coefficients of *pinth* DG and $\tilde{\gamma}_{s\tilde{k}}$ is a gain matrix. The reference signal for all the MG cluster system can be calculated in the tertiary layer integrating the error signals generated by the power flow between MGs as follows:

$$
\begin{cases}
\omega_s^{ref}(t_k^{T^*}) = \omega^{rated} + K_{s,pin}^P P_{s,pin}(t_k^{T^*}) \\
v_s^{ref}(t_k^{T^*}) = v^{rated} + K_{s,pin}^Q Q_{s,pin}(t_k^{T^*})
\end{cases}
\tag{30}
$$

The distribution secondary control (DSC) scheme was implemented using four control laws for frequency, voltage, active and reactive power corrections. To calculate the

frequency and voltage control laws, the relative outputs of neighboring terminals and a voltage observer ($\hat{v}_{s,i}$) are required. The frequency and voltage control laws can be described as:

$$
\begin{cases}
u_{s,i}^{\omega}(k+1) = \gamma_{i0}^{s} a_{i0}^{s} \left[ \omega_s^{ref}(t_k^{\tau^*}) - \omega_{s,i}(t_k^{\tau^*}) \right] + \sum_{j \in N_{s,i}} \gamma_{ij}^{s} a_{ij}^{s} \left[ \omega_{s,j}(t_k^{\tau^*}) - \omega_{s,i}(t_k^{\tau^*}) \right] \\[2mm]
u_{s,i}^{v}(k+1) = \gamma_{i0}^{s} a_{i0}^{s} \left[ v_s^{ref}(t_k^{\tau^*}) - \hat{v}_{s,i}(t_k^{\tau^*}) \right]
\end{cases}
\tag{31}
$$

For the active and reactive power sharing control laws design, two assumptions were considered. The first one was that the output power will be shared proportionally to the capacity of each DG's in steady state. The second one was that the active and reactive droop coefficients were chosen considering the maximum power capacities. Based on that, the consensus-based control laws can be expressed as:

$$
\begin{cases}
u_{s,i}^{P}(k+1) = \sum_{j \in N_{s,i}} \gamma_{ij}^{s} a_{ij}^{s} \left[ K_{s,j}^{P} P_{s,j}(t_k^{\tau^*}) - K_{s,i}^{P} P_{s,i}(t_k^{\tau^*}) \right] / K_{s,i}^{P} \\[2mm]
u_{s,i}^{Q}(k+1) = \sum_{j \in N_{s,i}} \gamma_{ij}^{s} a_{ij}^{s} \left[ K_{s,j}^{Q} P_{s,j}(t_k^{\tau^*}) - K_{s,i}^{Q} P_{s,i}(t_k^{\tau^*}) \right] / K_{s,i}^{Q}
\end{cases}
\tag{32}
$$

Having calculated all control laws in (31) and (32), the nominal frequency and voltage set points for each DG can be computed as:

$$
\begin{cases}
\omega_{s,i}^{nom}(t_k^{\ell+1}) = \omega_{s,i}^{nom}(t_k^{\ell}) + u_{s,i}^{\omega}(k) + K_{s,i}^{P} u_{s,i}^{P}(k) \\[2mm]
v_{s,i}^{nom}(t_k^{\ell+1}) = v_{s,i}^{nom}(t_k^{\ell}) + u_{s,i}^{v}(k) + K_{s,i}^{Q} u_{s,i}^{Q}(k)
\end{cases}
\tag{33}
$$

The calculated nominal values are sent to the primary control to regulate the frequency and voltage deviations. It is important to mention that the two-layer communication network was designed considering different time scales $(T_{sa}, \tau_{sa})$. Furthermore, to meet the time-scale separation criteria in the hierarchical information flow of the microgrids clusters, the dynamics of each layer were taken into account. In addition, using the Gershgorin circle theorem, the gain matrices $(\gamma_{ij}^{s}, \gamma_{i0}^{s}, \tilde{\gamma}_{s\tilde{k}})$ and terminal times $(\tau^*, T^*)$, can be calculated. Therefore, the distributed controllers have discrete inputs (e.g., neighbors' information and local measurements) that are updated every iteration, reducing the DG's communication bandwidth requirements (e.g., an intermittent low-bandwidth communication network with its neighbors). Figure 6 shows a graphical summary of the proposed approach.

Simulations were performed to evaluate the performance of the proposed scheme, considering the load change performance determination and plug-and-play assessment of the MG level for various communication network topologies. For the load change performance assessment, the proposed systems exhibited a robust behavior against link failures, communication delays, and data dropouts. During the data sharing between DGs or MGs, the system robustness was shown to be independent of the uncertainties intervals. For the plug-and-play evaluation, it was found that when the number of MGs increases the convergence time of the MG cluster increase, but the overall stability remains. Thus, the pinned DGs inside the MGs contribute to the tertiary communication task, and only pinned DGs' power flow discrepancy data must be transmitted in a distributed manner. This approach is different compared with conventional centralized tertiary schemes.

One important contribution of this work is the notorious reduction in the communication costs, since the discontinuous controllers are in a discrete mode. Moreover, each DG just partially needs the system specifications, perform local measurements, and finally intermittently communicate with its neighbors. A two-layer communication network considering several dynamics and time scales for each layer was proposed. In addition, an extensive stability analysis of the dynamics for the proposed system was presented.

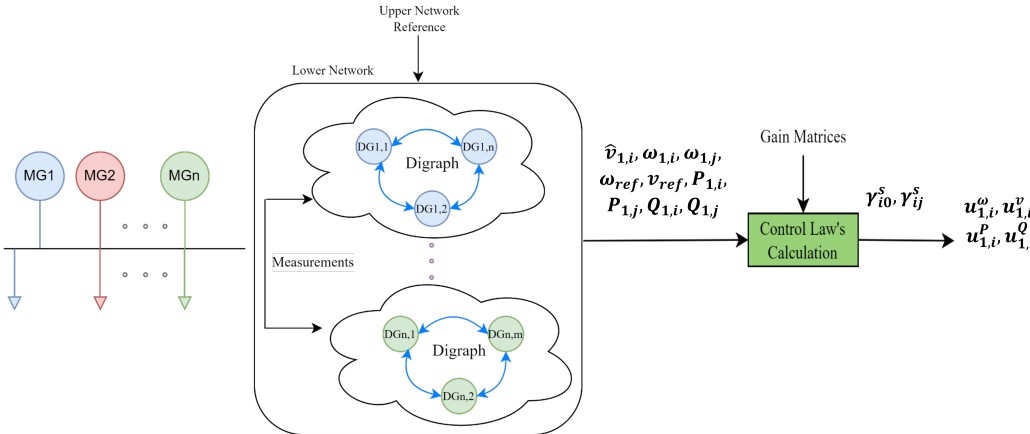

**Figure 6.** Implemented secondary-tertiary multi-layer communication approach for a cluster of MGs.

Similarly to the system proposed in [60], the authors in [66] proposed a distributed and networked secondary control method based on a two-layer scheme. For this approach the top layer is composed by an agent-based communication network,and the lower layer is the MG system form by different DGs and loads. Using the communication connections between both layers, agents can obtain information from the system and be able to transfer it to their neighbors using the communication network. As in [60], this research intends the regulation of voltage, frequency, and active and reactive power in islanded microgrids. For this purpose, the authors present a systematic method that allows deriving a set of control rules for the agents with fixed or dynamic weights, regardless of the type of communication network. It was accomplished using a unified approach that enables the algorithm's design for the management and control of microgrids changing one criteria in the control rules, using dynamic weights. The communication network used was the same presented in [34,39,41,44,45,57,60].

The design of the distributed and networked secondary control scheme assumed that the generation/demand relation remains in balance. In other words, the addition of the output change of controllable and uncontrollable DGs is equivalent to the total energy demand of the system during two successive time steps. Moreover, the number of agents must be the same as of DGs and loads. The agent's classification depends on the type of generator that is connected to the system. Therefore, if the directed digraph $G(V, E)$ is given, the control rules can expressed as:

$$B \cdot P(t) = W_P^T(t-1) \cdot [B \cdot P(t-1)] + U^T \cdot \Delta L^P - V^T \times [(I - B) \cdot \Delta P] \tag{34}$$

where $W_p$, $U$ and $V$ are the weighted matrices and these can be calculated as follows:

$$\begin{cases} V = D^{-1} \cdot A \\ U = (I - B) \cdot V + B \\ W_p(t-1) = I - diag(M_3 \cdot 1_{n \times 1}) + M_3 \end{cases} \tag{35}$$

From (35), it can be observed that the weighted matrices just require the neighbor's information and matrix $M_3$ to be calculated. The matrix $M_3$ is composed of sub-matrices and it can be computed as:

$$\begin{cases} M_1 = diag[P(t-1)] \cdot B \cdot (A - I) \\ M_2 = diag(X_p) \cdot 1_{n \times 1} \\ M_3 = \left| M_1 \circ M_2^{\circ(-1)} - (M_1 + M_1^T) \circ \left[ (M_2 + M_2^T)^{\circ(-1)} \right] \right| \end{cases} \tag{36}$$

where $1_{n \times 1}$ is a column vector, '$\circ$' and '$\circ(-1)$' represents the Hadamard product and Hadamard inverse function, respectively. The vector X is the parameter that can be replaced in the weighted matrices to change the outputs of the DGs through the control laws. Allowing the system to reach different targets iteratively. It is important to notice

that by replacing all the "*P*" and "*p*" terms for "*Q*" and "*q*" terms, respectively, the reactive power control law can be calculated. Figure 7 shows a graphical summary of the proposed approach.

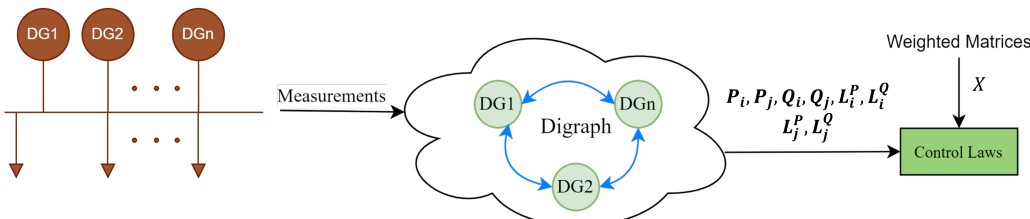

**Figure 7.** Implemented multi-layer communication strategy based on the vector parameter X to adjust the system's solution.

Several testing scenarios were used to assess the performance of the proposed scheme. These scenarios included the use of equal and proportional outputs for the controllable DGs and the study of the impact of different thresholds, topologies, link failures, and time delays on the system. Variable environmental conditions and load demands were considered to obtain a better view of the system performance. The most important outcome is that the active and reactive power amount of controllable DGs are almost the same along the entire simulation. In other words, the output of the controllable DGs quickly converged to a value that is proportional to their capacities. The variation in the obtained curves increased with the duration of the time delay. However, in a conventional approach, the changes of voltages takes more time when a longer time delay occurs, compared with the proposed approach. This is because the agents cannot receive the most current information to take effective corrective actions to follow the system changes. During iterations, the control rules with dynamic weights do not modify the system behavior, which means that the control rules maintain the supply–demand equilibrium.

The proposed scheme offers an integral and simple approach to achieve different management and control goals for MGs. The proposed systematic method provides the steps to acquire a group of control rules using dynamic and fixed weights. Those dynamic weighted laws redistribute the outputs of the DGs, so different targets can be achieved iteratively.

### *3.2. Event Trigger Approach*

This technique has been studied and implemented for different authors as [70–74]. The main idea is to provide robustness against communication problems using control schemes that reduce the system's dependency on the communication network, while performing an adequate voltage and frequency correction and assuring an accurate active and reactive (in some cases) power sharing. In [61], a secondary switched control approach for islanded microgrids that does not require a communication network was proposed. The proposed approach is based on a switched control technique that employs a time-dependent protocol to make the control system shift among two configurations. Thus, taking advantage of the characteristics of two different approaches. This approach just focuses on frequency restoration and active power sharing. Therefore, no voltage or reactive power regulation is performed. It is crucial to notice that even though the proposed secondary layer does not require a communication network, a generic communication system is still required for other microgrid functions.

The proposed method was designed to switch between a filtered proportional controller and an integral controller. Therefore, the control system in the Laplace domain can be represented as follows:

$$\delta = \begin{cases} \frac{k_i}{s + k_0 k_i} (\omega_0 - \omega), & k(t) > 0 \\ C, & k(t) = 0 \end{cases} \tag{37}$$

where $C$ is the filtered proportional controller constant value and $k(t)$ is the control parameter to be controlled by the time-dependent protocol. The first term of (37) represent the integral controller term, that is based on a low-pass filter with a cut-off frequency controlled by $k(t)$, assuming a constant value of $k_i$. It is important to mention that the values of $k_i$ and $k(t)$ affect the static and dynamic responses of the controller. In addition, the value of $k_0$ affects the trade-off between transient response time and accuracy. The proposed switching topology was designed to obtain the advantages of a low steady-state error while avoiding the drawbacks of the integral controller implementation.

In the proposed time-dependent protocol, $k(t)$ varies temporarily depending in the occurrence of an event. In other words, when an event is detected, the value of $k(t)$ varies instantaneously from zero to $k_{max}$ where it remains constant for a time frame $\Delta t$. After this time frame, the value of $k(t)$ linearly returns to zero. If another event is detected before the protocol procedure is completed, the time count and the initial conditions are reset, bringing $k(t)$ back to $k_{max}$, as depicted in Figure 8. Therefore, a good event detection system is critical to ensuring the performance of the suggested approach. Hence, two event-detection thresholds were used: one based on the proportion of active power change on the system and the other one in frequency changes. When at least one of the thresholds is exceeded, an event is identified.

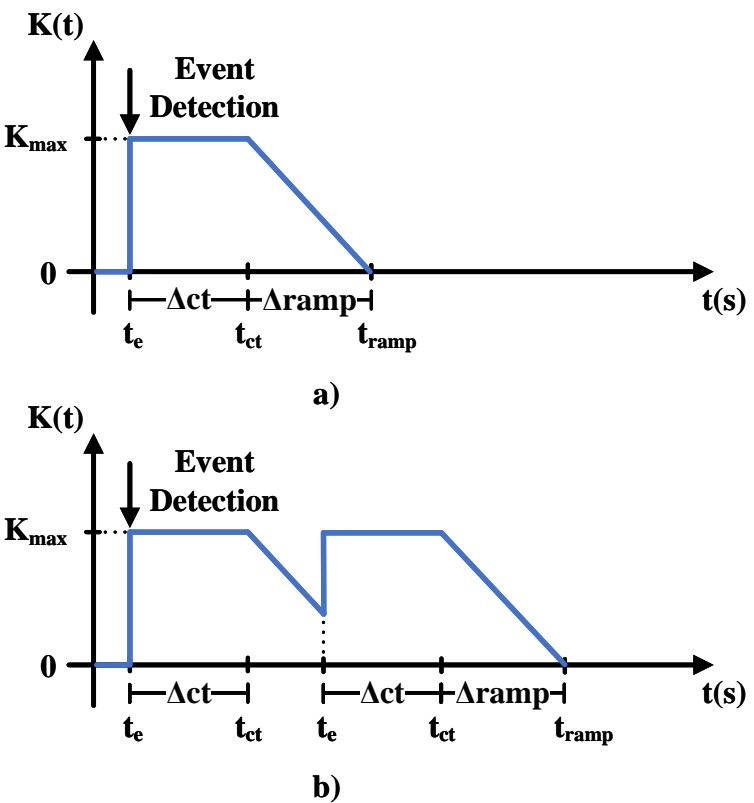

**Figure 8.** Time-dependent protocol for: (**a**) single event. (**b**) Multi-event scenarios.

The proposal of this secondary control system without a communication network for islanded microgrids is to offer a simple approach based on a switched control scheme and a time-dependent protocol. In addition to its simple design and implementation, this approach also has some other outstanding features. For instance, it offers improved flexibility and reliability, since no communications are required to operate the secondary control. Figure 9 shows a graphical summary of the proposed approach.

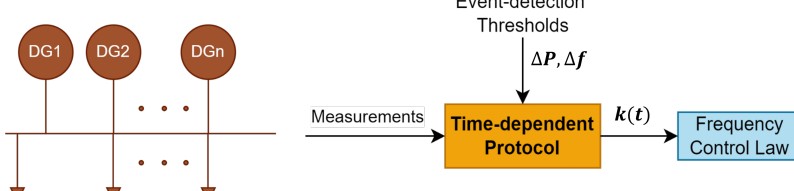

**Figure 9.** Diagram of the implemented secondary event-trigger controller based on generalized time-dependent protocol for singles or multiples events.

Several test setups were used to assess the performance of this scheme. First, the primary control layer was used to test the system, achieving an optimal power sharing. This was possible due to the proper design of the droop gains for each DG. Additionally, a frequency deviation was noted during the test, even though the power sharing was performed in a fast and accurate manner. For the second test, an invariant low-pass filter was used to change the value of the control parameter $K_0$. When it was high, the operation of the secondary layer was accurate and fast. However, it introduced a frequency error in the steady state. Conversely, when $K_0$ was low, the power sharing was slower, and a reduction in the frequency error was observed. The main contribution of the proposed scheme is that it simultaneously achieves both, a fast transient response, and an accurate steady state. An additional test was carried out to verify that the switched characteristics of the proposed scheme were enough to prevent the drawbacks of integral controllers. Finally, it was verified that for DGs with different droop gains and dynamically changing droops, an appropriate running frequency in steady and transient state was achieved.

In [62], a distributed secondary control strategy based on a decentralized event-triggered scheme was also considered. However, in this work a voltage regulation scheme was included. In this approach, a communication network is required for the secondary controllers used by the DGs. However, it is used only in particular moments. For this purpose, activating functions were developed to define the triggering times for different controllers, based on the occurrence an event. This strategy eliminates the recurrent communication dependency of the system, avoiding the high cost and inefficiencies caused by a large communication burden. For this work, the same communication network approach previously described was adopted in [34,39,41,44,45,57,60,66].

The proposed distributed control approach can be divided into two sub-controllers: focused on frequency and voltage restoration, respectively. Moreover, a virtual agent designated as "agent0" was used as the leader node to provide the references for the secondary controllers, assuming that the active power sharing can be established as a leaderless consensus problem. The frequency restoration sub-control is based on a redefined leader–follower consensus control strategy, that introduces the decentralized event-triggered approach and variable estimations. Therefore, the proposed leader-follower consensus distributed controller can be expressed as:

$$
\begin{cases}
u_{\omega i}(t) = & k_\omega e_{\omega i}(t) \\
e_{\omega i}(t) = & \sum_{j \in N_i} \left[ \hat{\omega}_j(t) - \hat{\omega}_i(t) \right] + d_i \left[ \omega_{ref} - \omega_i(t) \right]
\end{cases}
\tag{38}
$$

where $k_\omega$ is a constant greater than zero, $d_i$ is a binary number controlled by the communication between the *ith* DG and the agent0 and $e_{\omega i}(t)$ is the neighborhood tracking error. The proposed variation of the estimated values for the leader–follower consensus solution (denoted by $\wedge$), reduces the overall communication between agents. The estimation values can be defined as $\hat{\omega}_i(t) = \omega_i(t_k^{\omega i})$ with $t \in \left[ t_k^{\omega i}, t_{k+1}^{\omega i} \right)$. Therefore, estimation error can be written as:

$$
\varepsilon_{\omega i}(t) = \omega_i(t_k^{\omega i}) - \omega_i(t)
\tag{39}
$$

Equation (38) can ensure the global stability of the system frequency if at least one DG can always receive information from agent0 and the event-triggered time mechanism is defined as function of the neighborhood tracking error ($e_{\omega i}(t)$) and the estimation error ($\varepsilon_{\omega i}(t)$) as follows:

$$
\begin{cases}
t_k^{\omega i} = inf\left\{t > t_{k-1}^{\omega i}\,|\,f_{\omega i}(t) = 0\right\} \\[2ex]
f_{\omega i}(t) = \|\varepsilon_{\omega i}(t)\|^2 - \dfrac{\alpha_\omega\left(1 - \beta_\omega \sum_{j \in N_i} a_{ij} - \beta_\omega d_i/2\right)}{\sum_{j \in N_i} a_{ij}/\beta_\omega + d_i/(2\beta_\omega)}\|e_{\omega i}(t)\|^2
\end{cases}
\tag{40}
$$

where $\alpha_\omega$ and $\beta_\omega$ are values within an acceptable range. Using (39) and (40) an event-triggered time generation mechanism with a similar behavior as the one depicted on Figure 10 can be observed. Using this approach, the information exchange between generators only occurs at the event-triggered times. Otherwise, the communication between DGs it is not required. Generally speaking, the input values for the secondary control are estimated values, since the actual values are highly dependent on the communication network.

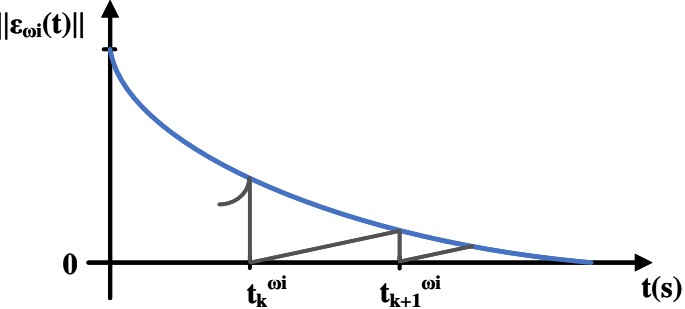

**Figure 10.** Event-triggered time generation mechanism.

Following the same logic, methodology and assumptions, the leader-follower consensus equations for the active power and active power estimation error can be written as:

$$
\begin{cases}
u_{pi}(t) = k_p e_{pi}(t) \\[1ex]
e_{pi}(t) = \displaystyle\sum_{j \in N_i} \left[D_{pj}\hat{P}_j(t) - D_{pi}\hat{P}_i(t)\right]
\end{cases}
\tag{41}
$$

where $k_p$ is a constant greater than zero, $D_p$ are the frequency droop control coefficients, and $p_i$ and $P_i$ are the output and filtered active power of each DG. The active power estimation error is described by:

$$
\varepsilon_{pi}(t) = D_{pi}P_i(t_k^{pi}) - D_{pi}P_i(t)
\tag{42}
$$

As in the distributed frequency controller, the global active power stability can be ensured by (41) if the event-triggered time meets the following:

$$
\begin{cases}
t_k^{pi} = inf\left\{t > t_{k-1}^{pi}\,|\,f_{pi}(t) = 0\right\} \\[2ex]
f_{pi}(t) = \|\varepsilon_{pi}(t)\|^2 - \dfrac{\alpha_p\left(1 - \beta_p \sum_{j \in N_i} a_{ij}\right)}{\sum_{j \in N_i} a_{ij}/\beta_p}\|e_{pi}(t)\|^2
\end{cases}
\tag{43}
$$

Based on the previous analysis, the control law for the frequency and active power controller can be derived by combining (38) and (41) as:

$$
\omega_{ni}(t) = \int\left[u_{\omega i}(t) + u_{pi}(t)\right]dt
\tag{44}
$$

For the leader–follower voltage controller the same methodology was applied. Similar results compared to (38) and (39) were obtained for the controller. In addition, similar event-triggered times and triggering function were obtained as follows:

$$
\begin{cases}
u_{vi}(t) = k_v e_{vi}(t) \\
e_{vi}(t) = \sum_{j \in N_i} \left[ \hat{V}_j(t) - \hat{V}_i(t) \right] + d_i \left[ V_{ref} - \hat{V}_i(t) \right]
\end{cases}
\tag{45}
$$

$$
\varepsilon_{vi}(t) = V_i(t_k^{vi}) - V_i(t)
\tag{46}
$$

$$
\begin{cases}
t_k^{vi} = inf\left\{ t > t_{k-1}^{vi} \,|\, f_{vi}(t) = 0 \right\} \\
f_{vi}(t) = \| \varepsilon_{vi}(t) \|^2 - \dfrac{\alpha_v \left( 1 - \beta_v \sum_{j \in N_i} a_{ij} - \beta_v d_i/2 \right)}{\sum_{j \in N_i} a_{ij}/\beta_v + d_i/(2\beta_v)} \| e_{vi}(t) \|^2
\end{cases}
\tag{47}
$$

Using the first term of (45), along with the voltage droop control, and the filtered reactive power equations, the voltage control law for this scheme can be expressed as:

$$
V_{ni}(t) = \int \left[ u_{vi}(t) + D_{qi}(\omega_c q_i - \omega_c Q_i) \right] dt
\tag{48}
$$

where $D_{qi}$ are the frequency droop control coefficients of the *ith* DG, $\omega_c$ is the cut-off frequency of the low-pass filter and $q_i$ and $Q_i$ are the output and filtered reactive power of each DG. Figure 11 shows a graphical summary of the proposed approach.

In this work, simulation results showed that the proposed distributed secondary control scheme can maintain an accurate active power sharing, while the frequency and the voltage are brought back to their normal values. The stability of the distributed control system can be assured due to the implemented event-triggered time system keeping away zero-behavior. This can be completed without adding any zero-like behavior. The event-triggered and sampling conditions for the active power, voltage, and frequency controllers, were defined through a novel distributed event-triggering rule. Allowing the authors to construct an easy-to-follow secondary control level. Due to the control implementation, the system can keep the stability against communication delays. The communication demand between the DGs secondary controllers was notoriously decreased, compared with the conventional communication scheme.

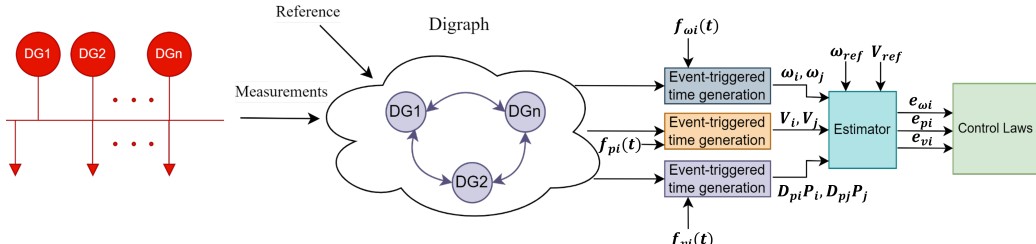

**Figure 11.** Implemented secondary event-triggered controller based on an multiple event-triggered mechanism.

All the communication robustness control strategies discussed are summarized in Table 2. It includes the technique used, active and reactive power sharing, voltage and frequency regulation, method used for stability analysis, communication network representation, and delays modeling.

**Table 2.** Summary of the recently published secondary control for power Communication Robustness.

| Concept | Power Quality Articles | | | | | | | | |
|---|---|---|---|---|---|---|---|---|---|
| | [59] | [60] | [61] | [62] | [63] | [64] | [29] | [65] | [66] |
| A | ✓ | ✓ | ✓ | ✓ | ✓ | ✓ | ✓ | ✓ | ✗ |
| B | ✗ | ✓ | ✗ | ✓ | ✓ | ✓ | ✓ | ✓ | ✗ |
| C | ✓ | ✓ | ✓ | ✓ | ✓ | ✗ | ✓ | ✓ | ✓ |
| D | ✗ | ✓ | ✗ | ✓ | ✓ | ✗ | ✗ | ✓ | ✓ |
| E | ✓ | ✓ | ✗ | ✓ | ✗ | ✓ | ✗ | ✓ | ✓ |
| F | ✓ | ✓ | ✗ | ✗ | ✗ | ✓ | ✗ | ✓ | ✗ |
| G | ✗ | ✓ | ✗ | ✗ | ✗ | ✗ | ✗ | ✗ | ✗ |
| H | ✗ | ✗ | ✓ | ✓ | ✗ | ✗ | ✗ | ✗ | ✗ |
| I | ✗ | ✗ | ✗ | ✓ | ✓ | ✓ | ✗ | ✗ | ✗ |
| J | ✗ | ✓ | ✗ | ✓ | ✗ | ✗ | ✗ | ✗ | ✗ |
| K | ✗ | ✗ | ✓ | ✗ | ✓ | ✗ | ✓ | ✗ | ✗ |
| L | ✗ | ✓ | ✓ | ✓ | ✓ | ✗ | ✗ | ✓ | ✗ |
| M | ✓ | ✗ | ✗ | ✗ | ✗ | ✓ | ✗ | ✗ | ✓ |
| N | ✗ | ✗ | ✗ | ✗ | ✗ | ✗ | ✓ | ✗ | ✗ |

Legend: A. Frequency regulation; B. Voltage Regulation; C. Active Power Regulation; D. Reactive Power Regulation; E. Graph modeling F. Delays modeling; G. Multi-layer method; H. Event-triggered method; I. Observer/Estimator implementation; J. Intermittent-communication approach; K. No-communication method; L. Lyapunov Stability; M. Lemma Stability; N. Root Locus stability.

## 4. Discussion

This paper has reviewed the control strategies for the improvement of power quality and robustness against problems in the communications network. There are different techniques and mathematical approaches that allow solving such problems in this type of scenario. In most cases, both approaches meet the general objectives of secondary control, providing voltage and frequency corrections, and moreover optimizing all the power sharing.

The authors of the reviewed papers in the state of the art work both parts (power quality and robustness against communication) with a greater emphasis on each of their topics. An important fact is that when implementing distributed controllers, the communication network tends to be represented as a digraph, which will be formed by agents that are responsible for interacting and sharing information between the different generators. Similarly, using optimization tools to develop controllers is an essential part for the development of secondary control techniques, that promotes the improvement of power quality and to correct related problems in the best way. This also generates flexibility at the time of design, since the constraints and conditions, as well as the methods implemented, will be the key to increase the controller performance.

On the other hand, for controllers that seek to add robustness to communication failures, it is notorious that the strategies need to work with estimates and predictions, making the dependence on actual measurements decrease greatly. This reduction in the dependence on real measurements is so low that authors such as [63] manage to achieve system operation without them, working adequately and efficiently using estimates. Although in the controllers with focus on power quality, approximations are also used, and the measured data are always necessary to make the appropriate corrections to the system. The examination of different applications and methods allows those places that suffered prolonged power outages when hit by natural disasters, implementing robust and updated MGs control strategies.

## 5. Conclusions

This paper presents a comprehensive overview of secondary control implementations on islanded microgrids. Different secondary control approaches that either focus on power-quality improvement or communication issues mitigation have been summarized. In addition, a comprehensive analysis of the implemented test setups and the reported results are presented. Most of the reviewed works do not use adequate reactive power sharing techniques and use the Lyapunov equation to study the stability of the implemented controllers. Although Lyapunov equation is an acceptable approach to assess the stability of a given controller, this method does not provide any information regarding the controller's behavior. Based on the presented analysis, it is evident that the voltage-reactive approach has not been explored yet. Finally, the network support functions have not been exploited on any secondary control scheme. Therefore, research efforts must be made to explore new secondary control approaches based on smart functions.

**Author Contributions:** Conceptualization, O.F.R.-M.; methodology O.F.R.-M., F.A. and A.C.L.; formal analysis, O.F.R.-M., C.A.V.-P., F.A. and A.C.L.; investigation, O.F.R.-M., F.A. and A.C.L.; resources, O.F.R.-M., F.A. and A.C.L.; writing—original draft preparation, O.F.R.-M. All authors have read and agreed to the published version of the manuscript.

**Funding:** This material is based upon work supported by the U.S. Department of Energy's Office of Energy Efficiency and Renewable Energy (EERE) under the Solar Energy Technologies Office Award Number DE-EE0002243-2144.

**Data Availability Statement:** Not applicable.

**Conflicts of Interest:** The authors declare no conflict of interest.

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
