# Peer review of "A Review of Distributed Secondary Control Architectures in Islanded-Inverter-Based Microgrids"

_energies, doi:10.3390/en16020878_

Round 1

Reviewer 1 Report

The author presents the article entitled “An Overview of Distributed Secondary Control Architectures in Islanded AC Microgrids”

This paper aims to ensure that all electrical system units correctly contribute to supplying the load in a pre-specified or optimized way. The Integrated distributed generators into the power system promote the integration and use of microgrids.

The article presents the following concerns:

  • Line 23: A reference is missing.

  • Line 39: Please check the citation style in the instruction for authors for grouped references. 

  • Line 96: Missing reference.

  • The objective of the manuscript must be restructured by highlightingh the novelty and the contributions of the work.

  • Tables 1 and 2 must be mentioned in the main text.

  • Line 372: Change “Fig 10” for “Figure 10”. 

  • Please add a Discussion section where the authors interpret the results obtained. 

  • Line 24 can be justified with the following grid definitions which consider the voltage, frequency and power: Transformerless common-mode current-source inverter grid-connected for pv applications; Leakage current reduction in single-phase grid-connected inverters - A review; Photovoltaic failure detection based on string-inverter voltage and current signals; Transformerless multilevel voltage-source inverter topology comparative study for PV systems; A new predictive control strategy for multilevel current-source inverter grid-connected; A novel integrated topology to interface electric vehicles and renewable energies with the grid.

My biggest concern is about the high level of coincidences. In its current form, the manuscript presents a 42%. Reference 64 is the work with most similitude. I suggest the authors to attend this observation to reduce the level of similitude.

The following misspelling should be checked:

  1. line 21: The article “A” may be incorrect. Consider changing it to agree with the beginning sound of the following word “MG”: “An MG”

  2. lines 39-40: Your sentence may be unclear or hard to follow. Consider rephrasing by this option: “This approach aims to ensure that all units correctly contribute to supply the load in a pre-specified or optimized way”

  3. line 45: The preposition “between” is incorrect here, change by “of”

  4. line 45: it seems that “type” may not agree in number with other words in this phrase. Change by: “Types”

  5. line 105: the adverbial “also” appears to be misplaced in this sentence. Determine the appropriate placement for the adverb. The correct form is: “also been”

  6. line 311: As the first word of the sentence, “this” should be capitalized.

  7. line 362: “In [60] a distributed…” It seems that you are missing a comma. Consider adding a comma: “In [60], a distributed…”

  8. line 407: It appears that the verb “exploit” should be in a participle form for the present perfect continuous tense. Consider changing the verb form by “exploiting” or “exploited”

Reviewer 2 Report

The paper makes a review of distributed secondary control architectures in Microgrids. The study enables and deals with solutions to current problems regarding the safe operation of microgrids.

The concise way in which the characteristics of the main control architectures used for the safe operation of microgrids are analyzed is highlighted.

The article has some aspects that should be reviewed before publication:

The authors must explicitly declare in the abstract and at the end of the introduction (before analysis) the contribution of the review concerning other reviews or declare if it is the first review to be carried out on distributed secondary control architectures in Microgrids.

A reference is missing on line 96.

Two sentences in line 311 do not begin with a capital letter.

Table 2, used in the conclusions, must be relocated. This table must be cited in the document.

Eliminate from the conclusions the texts that summarize the work carried out.

Two sentences in line 311 do not begin with a capital letter.

Table 2, used in the conclusions, must be relocated; this table must be cited in the document.

Eliminate from the conclusions the texts that summarize the work carried out.

Round 2

Reviewer 1 Report

Please, update the references as suggested.

Reviewer 2 Report

The authors took into account the recommendations given and the improvement is evident in the new version of the paper.